# Flow-based Image-to-Image Translation
# with Feature Disentanglement

**Ruho Kondo**
Toyota Central R&D Labs.
r-kondo@mosk.tytlabs.co.jp

**Keisuke Kawano**
Toyota Central R&D Labs.
kawano@mosk.tytlabs.co.jp

**Satoshi Koide**
Toyota Central R&D Labs.
koide@mosk.tytlabs.co.jp

**Takuro Kutsuna**
Toyota Central R&D Labs.
kutsuna@mosk.tytlabs.co.jp

## Abstract

Learning non-deterministic dynamics and intrinsic factors from images obtained through physical experiments is at the intersection of machine learning and material science. Disentangling the origins of uncertainties involved in microstructure growth, for example, is of great interest because future states vary due to thermal fluctuation and other environmental factors. To this end we propose a flow-based image-to-image model, called Flow U-Net with Squeeze modules (FUNS), that allows us to disentangle the features while retaining the ability to generate high-quality diverse images from condition images. Our model successfully captures probabilistic phenomena by incorporating a U-Net-like architecture into the flow-based model. In addition, our model automatically separates the diversity of target images into condition-dependent/independent parts. We demonstrate that the quality and diversity of the images generated for microstructure growth and CelebA datasets outperform existing variational generative models.

## 1 Introduction

Recently, machine learning models for generating various images conditioned on another image (diverse image-to-image models) have been developed [1, 2, 3, 4, 5] based on variational autoencoders (VAEs) [6] or generative adversarial networks (GANs) [7]. In the fields of material science, these models can be used for learning the relationship between the initial microstructure images and those after material processing. Figure 1 shows an example of microstructure growth, in which various microstructures ($x$) are obtained from an initial condition ($c$) via phase separation [8]. Such diversity is due not only to the elapsed processing time but to other environmental factors such as thermal fluctuation [8, 9, 10]. Our first goal is to model such a non-deterministic image translation, i.e., to generate diverse images ($x$) from the corresponding initial conditions ($c$).

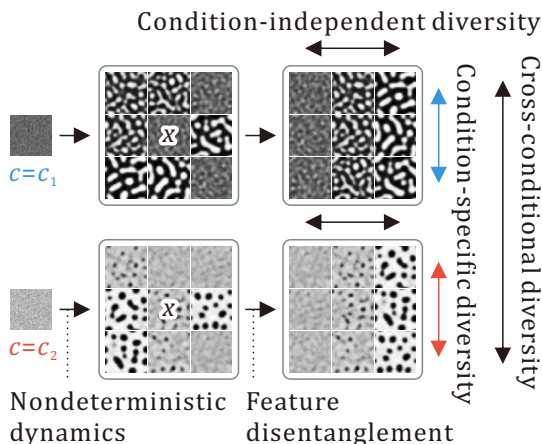

Figure 1: Illustration of our tasks: learning microstructure growth and disentangling its features. Condition and target are indicated as $c$ and $x$, respectively.

Our second goal is feature disentanglement. There are several origins of diversity within the above-mentioned image translations; some are from the given conditions (e.g., the mixture ratio of the two compounds) and the others are from environmental factors such as thermal fluctuation. For material property optimization [11], disentangling such diversity within those images is also important. Figure 1 illustrates our concept, which aims to disentangle the latent feature into (1) condition-dependent (referred to as "cross-conditional" and "condition-specific") and (2) condition-independent parts.

Flow-based generative models [12, 13, 14] are suitable when high-quality and diverse image generation with stable learning is required. Additionally, from the view point of application in downstream tasks such as property optimization [15] and inverse analysis [16], an invertible mapping between images and latent codes is a useful feature of flow-based models. However, to the best of the authors' knowledge, neither image-to-image translation nor feature disentanglement has yet been incorporated into flow-based models. With that point in mind, we propose a new flow-based image-to-image generative model and a novel technique that enables feature disentanglement in flow-based models. To condition a flow with an image, the prior distribution of the flow is associated with the condition image through a multi-scale encoder. To disentangle latent features into condition-specific and condition-invariant parts, we force a part of the outputs of the encoder to be independent of the condition image.

Our contributions can be summarized as follows:

- We propose a diverse image-to-image translation model based on flow that outperforms current state-of-the-art variational models in terms of the Fréchet Inception Distance (FID) [17] and Learned Perceptual Image Patch Similarity (LPIPS) [18] for the CelebFaces Attributes (CelebA) [19] dataset and our original microstructure dataset created by solving the Cahn–Hilliard–Cook (CHC) equation [8].
- A method to incorporate feature disentanglement mechanism into flow-based generative models is proposed. The proposed method successfully separates embedded features into the condition-dependent and condition-independent parts.

The remainder of this paper is organized as follows. Section 2 states the preliminaries and Section 3 outlines our proposed model. Section 4 describes related work. Section 5 shows the results of image generation using the CelebA [19] and CHC datasets. Section 6 summarizes and concludes our study.

## 2 Preliminaries

**Flow with multi-scale architecture**  In flow-based generative models [12, 13, 14], the data distribution and the latent distribution are connected by an invertible mapping. The map $f_\theta : x \mapsto z$ is commonly expressed by a neural network and is trained to maximize the likelihood of data points pushed-forward from the data distribution measured on the latent distribution. Here, $\theta$, $x$ and $z$ are the parameter, target image, and latent vector, respectively. Flow-based models simultaneously achieve high representation power and numerically stable optimization because they do not require approximation, as do VAEs [6], nor a discriminator, as in GANs [7]. For high-dimensional data generation, the use of a multi-scale architecture [13] is preferred. The spatial resolution of the feature map is recursively reduced to half as the level of flow increases. In the generation phase, $x$ is obtained as

$$\xi^{(L-1)} = g^{(L)}(z^{(L)}), \ \ \xi^{(l)} = g^{(l+1)}(z^{(l+1)} \oplus \xi^{(l+1)}), \ \ x = \xi^{(0)}, \ \ l = 0, \cdots, L-2, \qquad (1)$$

where $g^{(l)} = (f^{(l)})^{-1}$, $f^{(l)}$ is the invertible map at level $l$, $\oplus$ is concatenation along the channel dimension, and $L$ is the total level of flow. For unconditional generation, $z^{(l)}$ is obtained by $z^{(l)} \sim \mathcal{N}(\mathbf{0}, \mathbf{1})$.

**U-Net**  U-Net [20] is one of the most popular architectures for image-to-image translation models that comprise encoder-decoder networks [21] and skip paths. When input image $c$ is being encoded into the latent vector, skip paths are created when the image resolution is halved. The spatial resolution of $c$ is reduced $L$ times during encoding as follows:

$$u_c^{(0)} = c, \ \ u_c^{(l)} = \text{NN}(u_c^{(l-1)}), \ \ l = 1, \cdots, L, \qquad (2)$$

where NN is a neural network. The target image $x$ is generated from $u_c^{(l)}$ as $\xi_c^{(L-1)} = \text{NN}(u_c^{(L)})$, $\xi_c^{(l)} = \text{NN}(u_c^{(l+1)} \oplus \xi_c^{(l+1)})$, $x = \xi_c^{(0)}$, $l = 0, \cdots, L-2$, which is very similar to Eq. (1).

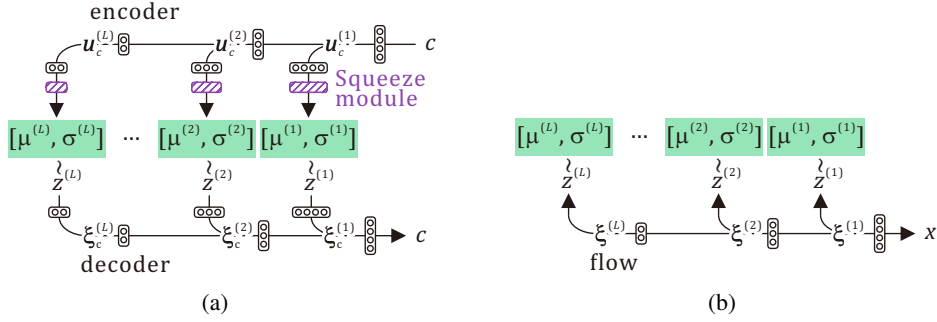

Figure 2: Overview of our proposed Flow U-Net with Squeeze modules (FUNS), which consists of the following two parts that are trained simultaneously in an end-to-end manner. (a) U-Net-like variational autoencoder that encodes condition $c$ to latent feature $z$. Purple hatched boxes represent the squeeze modules, which disentangle the latent feature. (b) Flow-based generative model with multi-scale architecture that learns an invertible mapping between target image $x$ and latent feature $z$.

## 3 Proposed Model

An overview of our proposed Flow U-Net with Squeeze modules (FUNS) is shown in Fig. 2. The model consists of a U-Net-like variational autoencoder with multiple scales of latent variables between its encoder and its decoder. These latent spaces encode an image $c \in \mathbb{R}^{d_c}$ to condition on, which is trained by reconstructing $c$. Simultaneously a flow-based model is trained to associate possible outputs $x \in \mathbb{R}^{d_x}$ with the given image $c$ by maximizing the likelihood of the latents $z \in \mathbb{R}^{d_x}$ that the flow produces given $x$ under the U-Net's latent space densities for $c$ (see Section 3.1).

Feature disentanglement is achieved as follows. Let us consider a decomposition of the following form: $p(z|c) = p(z^{\text{iv}})p(z^{\text{sp}}|c)$, which orthogonally decomposes the latent vector into condition independent and dependent parts (see Section 3.2). Here, $z^{\text{sp}}$ can be considered as a random variable that should be almost uniquely determined under a given condition $c$. For this purpose, we introduce the entropy regularization to narrow $p(z^{\text{sp}}|c)$ (see Section 3.3).

### 3.1 Flow U–Net

The outputs of the encoder part of U-Net and the inputs of the flow architecture both have multi-scale architectures, as can be seen in Eq. (1) and (2). This allows us to consistently plug the encoder outputs of U-Net ($u_c^{(l)}$) into the flow inputs, $z^{(l)}$, as follows (see Fig. 2):

$$z^{(l)} \sim \mathcal{N}(\mu^{(l)}(c), \text{diag } \sigma^{(l)2}(c)), \;\; [\mu^{(l)}(c), \log \sigma^{(l)2}(c)] = \mathcal{F}^{(l)}\left(u_c^{(l)}\right), \;\; l = 1, \cdots, L \quad (3)$$

where $\mathcal{F}^{(l)}$ is an arbitrary function. In this study, we use the squeeze module for the $\mathcal{F}^{(l)}$ proposed in the next section. As can be seen in the above equation, $z^{(l)}$ is now sampled from the distribution that depends on $c$. This achieves the multi-scale conditional sampling of the latent vector for the flow. In what follows, we abbreviate the superscript $(\bullet)^{(l)}$ and write $e_\phi(z|c) \equiv \mathcal{N}(\mu^{(1)} \oplus \cdots \oplus \mu^{(L)}, \text{diag } (\sigma^{(1)2} \oplus \cdots \oplus \sigma^{(L)2}))$ for simplicity, where $\phi$ is a parameter corresponding to $\mathcal{F}$ and $u_c$.

We now consider the use of a decoder whose output distribution is $d_\psi(c|z)$, with a parameter $\psi$ (as described in Fig. 2(b)). The decoder is used to reconstruct $c$ from $z$, where $z$ is drawn from $e_\phi(z|c)$. The addition of the decoder results from the need to maximize the mutual information [22]. More specifically, the mutual information between $z$ and $c$, $I(z; c)$, is increased by minimizing the reconstruction error (see the Supplementary Material). A normal distribution with unit variance is used for $d_\psi(c|z)$. We call the flow with the multi-scale encoder/decoder a Flow U-Net. Very recently, a similar conditional flow model that also contains an encoder and decoder was proposed [23]. The main differences between their model and ours are that our model (i) considers multi-scale conditioning and (ii) contains squeeze modules.

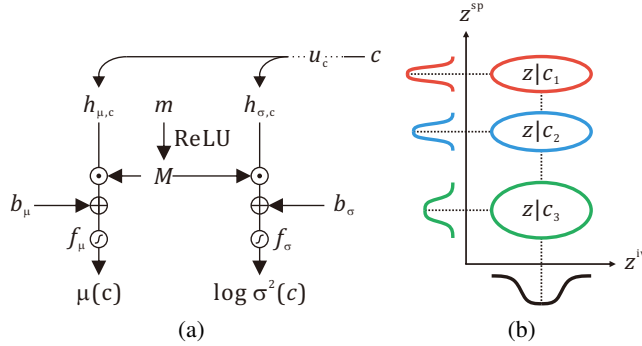

|     |     |
| --- | --- |
| (a) | (b) |

Figure 3: Detail of squeeze module. Here, $\odot$ and $\oplus$ are element-wise multiplication and element-wise addition, respectively.

## 3.2 Disentangling Latent Features Using Squeeze Modules

Disentangling the latent vector $z$ into condition-specific and condition-invariant parts cannot be achieved by Flow U-Net because the latent vector is fully conditioned on $c$. To separate $z$ into these parts, we introduce a *squeeze module* (Fig. 3(a)), with which the actual dimensionality of $z$ conditioned on $c$ is squeezed without changing the total dimensionality of $z$. This is required for flow, in which both $x$ and $z$ must have the same dimensions. Because the squeezed dimension of $z$ no longer depends on $c$, it captures the features that are common among the conditions.

It is well known that one can easily obtain a sparse latent vector by applying a rectifying nonlinear function and minimizing the $L_1$ norm of its outputs [24]. By using this technique, we make a part of latent vector $z$ follow $\mathcal{N}(f_\mu(b_\mu), \mathrm{diag}(\exp(f_\sigma(b_\sigma))))$ that is independent of $c$. To accomplish this, the following module is used for the $\mathcal{F}$ that appears in Eq. (3).

$$M = \mathrm{ReLU}(m), \qquad\qquad [h_{\mu,c}, h_{\sigma,c}] = \mathrm{NN}(u_c),$$
$$\mu(c) = f_\mu(M \odot h_{\mu,c} + b_\mu), \qquad\qquad \log \sigma^2(c) = f_\sigma(M \odot h_{\sigma,c} + b_\sigma)$$

Here, $m$, $b_\mu$, and $b_\sigma$ are, respectively, the learnable parameters whose dimensions are equal to that of $\mu(c)$, the neural networks (NN), and the rectified linear unit (ReLU). In addition, $\odot$ is the element-wise product. The initial value of $m$ is set to $m \sim \mathcal{N}(0, 1)$. Functions $f_\mu(\bullet) = a \tanh(\bullet)$ and $f_\sigma(\bullet) = b(\exp(-(\bullet)^2) - 1)$ are the activation functions that respectively restrict $\mu(c) \in (-a, a)$ and $\log \sigma^2(c) \in (-b, 0]$, where $a$ and $b$ are hyperparameters. Limiting the value of $\mu(c)$ and $\log \sigma^2(c)$ is only done to facilitate numerical stability. In our experiments, we use $a = 2$ and $b = \log(2\pi e)$. When $M_i = 0$ where $(\bullet)_i$ indicates the $i$-th element of $(\bullet)$, both $\mu(c)_i$ and $\log \sigma^2(c)_i$ are independent of $c$. In what follows, $z^{\mathrm{sp}}$ and $z^{\mathrm{iv}}$ are defined as the parts of $z$ where $M_i > 0$ and $M_i = 0$, respectively. Figure 3(b) illustrates the relationship between $z^{\mathrm{sp}}$ and $z^{\mathrm{iv}}$. The distributions of $z^{\mathrm{sp}}$ depends on $c$ whereas those of $z^{\mathrm{iv}}$ do not.

Note that modeling the relation between $z^{\mathrm{sp}}$ and $c$ as a probabilistic relation is important because, as we will see in the experiments, the generated images may be diverse under a specific condition $c$ but may not be under another condition $c'$. In addition, marginalizing $p(z^{\mathrm{sp}}|c)$ with respect to $c$ using $p(c)$, we obtain $p(z^{\mathrm{sp}})$, which corresponds to the "cross-conditional diversity" in Fig. 1.

## 3.3 Loss Function

The loss functions of the conditional flow $\mathcal{L}^{\mathrm{flow}}$ and those of the condition reconstruction $\mathcal{L}^{\mathrm{recons}}$ are respectively written as follows:

$$\mathcal{L}^{\mathrm{flow}}(\theta, \phi, M) = \mathbb{E}_{x,c \sim p(x,c)}\Big[ -\log e_\phi(f_\theta(x)|c) - \sum_p \log J_p \Big], \qquad (4)$$

$$\mathcal{L}^{\mathrm{recons}}(\phi, \psi, M) = \mathbb{E}_{c \sim p(c),\, z \sim e_\phi(z|c)}\big[ -\log d_\psi(c|z) \big], \qquad (5)$$

where $J_p$ is the Jacobian of the $p$-th invertible mapping. Moreover, $\mathcal{L}^{\mathrm{flow}} + \mathcal{L}^{\mathrm{recons}}$ is an upper bound of $\mathbb{E}_{x,c \sim p(x,c)}\big[ -\log p_{\theta,\phi}(x|c) \big] - I(c; z)$, which is an objective for conditional generation (see the

Supplementary Material). The squeezing loss function $\mathcal{L}^{\text{squeeze}}$ is an $L_1$ loss of $M$ because the partial dimensions of mask $M$ are required to be zero. That is,

$$\mathcal{L}^{\text{squeeze}}(M) = \|M\|_1. \tag{6}$$

Moreover, to reduce the uncertainty of the $z^{\text{sp}}$ given $c$, the entropy of $z^{\text{sp}}$ conditioned on $c$ (a part of $z$ where $M_i > 0$) should be decreased. Hence, the following entropy regularization is introduced.

$$\mathcal{L}^{\text{entropy}}(\phi, M) = \sum_{i=1}^{d_x} M_i \mathcal{H}[e_\phi(z_i|c)], \tag{7}$$

where $\mathcal{H}[\bullet]$ is the entropy and $e_\phi(z|c) = \prod_i e_\phi(z_i|c)$ is assumed. When $e_\phi(z_i|c)$ is the normal distribution with standard deviation $\sigma_i$, it can be written as $\mathcal{L}^{\text{entropy}} = \frac{1}{2} \sum_{i=1}^{d_x} M_i \big( \log \sigma_i^2 + \log(2\pi e) \big)$. It is noteworthy to mention that, without the entropy regularization, we fail to disentangle the features (see the experiments and Fig. 7). In summary, the total loss for FUNS is as follows: $\mathcal{L}^{\text{FUNS}} = \mathcal{L}^{\text{flow}} + \mathcal{L}^{\text{recons}} + \alpha \mathcal{L}^{\text{squeeze}} + \beta \mathcal{L}^{\text{entropy}}$, where $\alpha$ and $\beta$ are hyperparameters that control the amount of regularization.

### 3.4 Sampling Procedure

For conditional generation, $\mu(c)$ and $\log \sigma^2(c)$ are first obtained by passing condition $c$ through the encoder. The latent vectors $z$ conditioned on $c$ are sampled from $\mathcal{N}(\mu(c), \text{diag } \sigma^2(c))$ and then, samples $x$ are obtained as $x = f_\theta^{-1}(z)$. In addition to ordinary conditional generation, our model can also generate condition-specific diverse samples $x^{\text{sp}}$ and condition-invariant diverse samples $x^{\text{iv}}$ from a given sample $x^{\text{given}}$. Such condition-specific and condition-invariant diverse samples are those whose diversities originate from conditions and other sources, respectively. Samples $x^{\text{sp}}$ and $x^{\text{iv}}$ can easily be obtained by resampling $z^{\text{sp}}$ and $z^{\text{iv}}$, respectively, even if the ground truth condition is unknown. Moreover, from two given images, $x_1^{\text{given}}$ and $x_2^{\text{given}}$, style transferred samples $x_{1,2}^{\text{trans}}$ and $x_{2,1}^{\text{trans}}$ can be obtained by exchanging their condition-invariant features. The sampling procedures used to obtain the above mentioned samples are summarized in Algorithm 1.

## 4 Related Work

**Conditional Flows**  Conditional flow models [14, 16, 23] are categorized into two classes. The first separates latent vector $z$ into condition-specific and condition-invariant parts in advance and uses $c$ as the former [16]. This type of model, however, can only be used when the dimensionality of $c$ is less than or equal to that of $x$ because of the flow restriction requiring that the dimensionalities of $x$ and $z$ be the same. In the other class, a part of $z$ is encoded from $c$ [14] or all of $z$ is encoded from both the $c$ and the noise [23]. This type of model can potentially be used to treat a $c$ whose dimensionality is much larger than that of $x$ because the encoder can reduce the dimensionality of $c$. The Flow U-Net described in Section 3.1 is classified as this type. However, to separate $z$ into $z^{\text{sp}}$ and $z^{\text{iv}}$, it is necessary to choose those dimensions carefully. Generally speaking, before the start of model

---

**Algorithm 1** Procedures used to obtain condition-specific diverse images $x^{\text{sp}}$, condition-invariant diverse images $x^{\text{iv}}$, and style transformed images $x^{\text{trans}}$. $(\bullet)_{\text{mean}}$ is the mean of $(\bullet)$.

| **Require:** $x^{\text{given}}$ | **Require:** $x^{\text{given}}$ | **Require:** $x_1^{\text{given}}, x_2^{\text{given}}$ |
|---|---|---|
| $z = f_\theta(x^{\text{given}})$ | $z = f_\theta(x^{\text{given}})$ | $z_1 = f_\theta(x_1^{\text{given}})$ |
| $c' = (d_\psi(c|z = z))_{\text{mean}}$ | $\mu = f_\mu(b_\mu),\ \log \sigma^2 = f_\sigma(b_\sigma)$ | $z_2 = f_\theta(x_2^{\text{given}})$ |
| $z' \sim e_\phi(z|c = c')$ | $z' \sim \mathcal{N}(\mu, \text{diag } \sigma^2)$ | **for** $i = 1$ to $d_x$ **do** |
| **for** $i = 1$ to $d_x$ **do** | **for** $i = 1$ to $d_x$ **do** |   **if** $M_i = 0$ **then** |
|   **if** $M_i > 0$ **then** |   **if** $M_i = 0$ **then** |     $z_{1,i} \leftarrow z_{2,i}$ |
|     $z_i \leftarrow z_i'$ |     $z_i \leftarrow z_i'$ | $x_{1,2}^{\text{trans}} \leftarrow f_\theta^{-1}(z_1)$ |
| $x^{\text{sp}} \leftarrow f_\theta^{-1}(z)$ | $x^{\text{iv}} \leftarrow f_\theta^{-1}(z)$ | **return** $x_{1,2}^{\text{trans}}$ |
| **return** $x^{\text{sp}}$ | **return** $x^{\text{iv}}$ | |

---

training, there is no oracle to indicate how large the dimensionality of $z$ conditioned on $c$ should be for conditional generation. This becomes a crucial problem when $c$ consists of high-dimensional data, such as in the case of images. In FUNS, the dimension of $z^{\text{sp}}$ is learnt from the data.

**Conditional Variational Models**   Variational approaches for image generation with feature disentanglement and diversity have progressed recently. In conditional variational autoencoders (CVAEs) [25], the latent vector $z$ is separated into two parts, condition and noise parts, in advance. Variational U-Net (VUNet) [1] and Probabilistic U-Net (PUNet) [2] employ similar strategies to generate diverse images. In fact, all of the variational models use a loss function that is similar to that used in our model (see the Supplementary Material), except for the following differences: (i) our model does not use approximated distributions; (ii) it has no deterministic path from condition $c$ to target $x$; and (iii) our model uses invertible neural networks for generating $x$ from $z$.

**Conditional Generative Adversarial Networks**   BicycleGAN [3] is a diverse image-to-image generative model based on conditional GANs [26]. Similar to CVAE, BicycleGAN generates $x$ from both $c$ and noise. However, unlike CVAE, the BicycleGAN generator is trained to fool the discriminator, whereas the discriminator is trained to distinguish the generated and real images. In addition, cd-GAN [27] and cross-domain disentanglement networks [28] are both extended models that enable feature disentanglement. MUNIT [4] and DRIT [5] are both extended models that learn from unpaired data.

## 5   Experiments

**Baselines**   We compared our model to the official implementations of VUNet [1] and PUNet [2]. VUNet was found to work well when the coefficient of the Kullback-Leibler divergence loss was fixed to one throughout the training, whereas in PUNet, the prediction loss function was changed from the softmax cross-entropy to mean squared error to predict real-valued images.

**Datasets**   We employed the celebrity face attributes dataset (CelebA [19], which consists of images with 40 attribute annotations) as well as our orginal dataset, Cahn–Hilliard–Cook (CHC) dataset. The CHC dataset includes a vast number of microstructure images describing phase separation and was created by solving the following partial differential equation: $\frac{\partial u}{\partial t} = \nabla^2(u^3 - u - \gamma\nabla^2 u) + \sigma\,\zeta$, where $u$ is the local density, $t$ is the time, $\gamma$ is the material property, $\sigma$ is the intensity of noise, and $\zeta \sim U([-1,1])$. Here, $\gamma = 1.5$ and $\sigma = 2$ are used. We generated 999 distinct initial conditions ($c$ in Fig. 2(a)). For each initial condition, we solved the CHC equation 32 times to obtain $u(t)$ for $0 \leq t \leq 9,000\Delta t$ where $\Delta t = 0.01$ is the time increment, which varies depending on the noise $\zeta$. Of these, 800 trajectories were used for training, 100 were used for validation and 99 were used for testing. From these trajectories, we created the CHC dataset as follows: $(c, x) \equiv (u(0), u(1,000k\Delta t))$ where $k = 1, 2, \cdots, 9$. In total, the CHC dataset includes 287,712 image pairs. The number of data for training, validation and testing in CelebA were 162,770, 1,9867 and 19,962, respectively, which follows an official train/val/test partitions. The CelebA image sizes were reduced to $64 \times 64$, and the CHC image sizes were also $64 \times 64$. In the training phase, CelebA color depth was reduced to five bits following the Glow settings [14], whereas CHC was maintained at eight bits. Random noise smaller than the color tone step was added to CelebA, in a process known as dequantization [29]. The data were then rescaled to be in the range $[-1, 1]^{d_x}$. Because there are no condition images in CelebA, one image in each tag was chosen as the condition image and all images in that tag were assumed to be conditioned on those selected images. The "Smiling" attribute was chosen for the CelebA tag. One Smiling image $c^{\text{smile}}$ and one non-Smiling image $c^{\text{not smile}}$ were chosen from the training data for condition images and were fixed during training and testing. All of the Smiling images were conditioned on $c^{\text{smile}}$, whereas all of the non-Smiling images were conditioned on $c^{\text{not smile}}$.

**Implementation Details**   We implemented our model with Tensor Flow version 1.10.0 [30]. We used Glow [14] for the flow architecture, where the number of flows per level $K$ and the total levels $L$ were $K = 48$ and $L = 4$, respectively. In all of our experiments, $\alpha = 0.01$ and $\beta = 0.1$ were used. The encoder and decoder had mirror symmetric architectures that were consistent with $3L$ residual blocks, where each block contains two batch normalizations [31], two leaky ReLUs and two convolutions with a filter size of three. The image size was halved for every three residual blocks in the encoder and was doubled for every three residual blocks in the decoder. The number of

| Condition | Ground truth | VUNet [1] | PUNet [2] | FUNS (ours) |
|---|---|---|---|---|

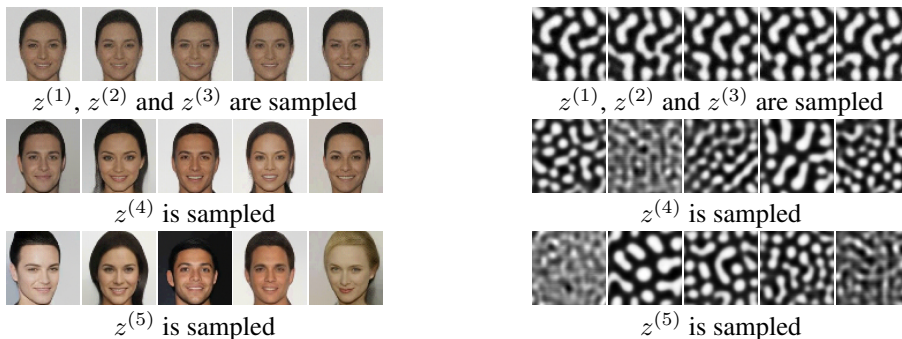

<center>Figure 4: Comparison of diverse image generation with various models.</center>

$z^{(1)}$, $z^{(2)}$ and $z^{(3)}$ are sampled

$z^{(1)}$, $z^{(2)}$ and $z^{(3)}$ are sampled

$z^{(4)}$ is sampled

$z^{(4)}$ is sampled

$z^{(5)}$ is sampled

$z^{(5)}$ is sampled

<center>Figure 5: Multi-scale samples for CelebA and CHC.</center>

convolution channels was 16 in the first layer for the encoder and doubled when the image size was halved. For training, we used the default settings of the Adam [32] optimizer with a learning rate of $10^{-4}$ and batch size of 32. All experiments were carried out on a single NVIDIA Tesla P100 GPU.

**Evaluation Metrics** Image quality was measured by FID [17] using the Inception-v3 model [33]. Note that a smaller FID implies better image quality. To evaluate the FID of CelebA and CHC, 50,000 images per class and 3,168 images, respectively, were used. Image diversity was measured by LPIPS [18]. This is the sample-wise distance between activations obtained by putting the samples into the pretrained model. In this study, we used a modified version of this measure because LPIPS is too sensitive for inter-conditional diversity. Namely, the LPIPS between images obtained under different conditions is greater than that between images obtained under the same conditions. We also measure the intra-conditional diversity to prevent the overestimation of LPIPS values caused by the diversity of conditions. Accordingly, let LPIPS $(c, c')$ be an LPIPS between images generated from condition $c$ and $c'$. The intra-conditional LPIPS ($c$-LPIPS) is defined as $\mathbb{E}_{c \sim p(c)} \big[ \text{LPIPS}(c, c) \big]$. The numbers of images used for evaluating both LPIPS and $c$-LPIPS were 4,000 for CelebA and 3,168 for CHC. To ensure that the generated images were conditioned on $c$, the prediction performance of $c$ from a generated $x$ was measured by in-house ResNet that was trained on real data. For CelebA, the prediction model was trained to classify whether the input image belongs to Smile or not and its accuracy was measured. For CHC, the prediction model was trained to predict condition image $c$ and the $L^2$ distance between the ground truth $c$ and the predicted one was measured.

$x^{\text{given}}$

$x^{\text{sp}}$

$x^{\text{iv}}$

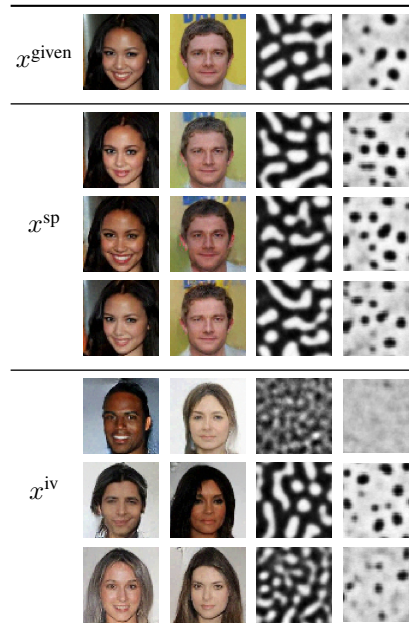

Figure 6: Condition-specific and condition-invariant diverse samples $x^{\text{sp}}$ and $x^{\text{iv}}$, respectively, obtained from the given image $x^{\text{given}}$ (test image).

Table 1: Comparison of FID, LPIPS and $c$–LPIPS. The means and standard deviations of five trials are shown for FID. All of the standard deviations of five trials for LPIPS and $c$-LPIPS are less than 0.021 (abbreviated). Full information can be found in the Supplementary Material.

| | | CelebA | | | CHC | | |
|---|---|---|---|---|---|---|---|
| | | FID | LPIPS | $c$–LPIPS | FID | LPIPS | $c$–LPIPS |
| VUNet | $T = 1.0$ | $66.0 \pm 4.3$ | 0.148 | 0.146 | $96.5 \pm 2.1$ | 0.217 | 0.118 |
| VUNet | $T = 0.8$ | $81.7 \pm 3.7$ | 0.105 | 0.103 | $164.0 \pm 5.7$ | **0.249** | 0.113 |
| PUNet | $T = 1.0$ | $114.8 \pm 9.2$ | 0.182 | 0.180 | $225.7 \pm 6.1$ | 0.226 | 0.108 |
| PUNet | $T = 0.8$ | $117.2 \pm 5.5$ | 0.149 | 0.146 | $227.9 \pm 6.0$ | 0.214 | 0.088 |
| FUNS | $T = 1.0$ | $39.6 \pm 3.9$ | **0.264** | **0.262** | **$10.5 \pm 2.0$** | 0.207 | **0.157** |
| FUNS | $T = 0.8$ | **$29.5 \pm 3.5$** | 0.259 | 0.256 | $11.1 \pm 2.5$ | 0.210 | 0.155 |
| Real data | | – | 0.286 | 0.284 | – | 0.225 | 0.169 |

Table 2: Prediction performance of generated images with respect to their ground truth condition. For CelebA, classification accuracy (Acc.), i.e., whether the generated images belong to the Smile class or not is measured. For CHC, the $L^2$ distance (Err.) between the predicted $c$ and ground truth is measured.

| | | Acc. | Err. |
|---|---|---|---|
| VUNet | $T = 1.0$ | 0.977 | 9.46 |
| VUNet | $T = 0.8$ | 0.981 | **9.44** |
| PUNet | $T = 1.0$ | **1.000** | 9.48 |
| PUNet | $T = 0.8$ | **1.000** | 9.48 |
| FUNS | $T = 1.0$ | 0.973 | 9.67 |
| FUNS | $T = 0.8$ | 0.967 | 9.81 |
| Real data | | 0.924 | 9.54 |

**Results** Figure 4 shows images generated by various methods. To generate high-quality images, all of the images were sampled from the reduced-temperature model $p_{\text{model},T}(x|c) \propto (p_{\text{model}}(x|c))^{T^2}$ [34] where $p_{\text{model}}$ is a model distribution and $T$ is the sampling temperature. Here, $T = 1.0$ is used for the all cases except for CelebA with FUNS. In that case, $T = 0.8$ is used, because it yields the best performance in terms of FID (see Table 1). PUNet tends to generate blurred images, whereas VUNet and FUNS successfully generate sharp images. It appears that all of these models can generate diverse images. Additional samples are shown in the Supplementary Material.

As noted in Eq. (3), there are several levels of latent features, $z^{(1)}, \cdots , z^{(L)}$, in our model. Figure 5 shows generated samples for which only a part of $z^{(l)}$ are sampled whereas other latent features are fixed to $\mu^{(l)}(c)$. We can see that a larger diversity is captured by latent features at lower resolution levels ($z^{(4)}$ and $z^{(5)}$) whereas very subtle variations are captured by higher resolution levels ($z^{(1)}$, $z^{(2)}$ and $z^{(3)}$).

The quantitative comparisons of the models are summarized in Table 1, where it can be seen that our model outperforms the competitors in terms of both FID and $c$-LPIPS for all datasets. Note that for the CHC dataset, LPIPS has a much larger value than $c$-LPIPS. This means that the original LPIPS overestimates the diversity because of the inter-conditional diversity. Table 2 summarizes the prediction performance of the conditions from the generated data. Note that the prediction error for CHC is 22.84 when the ground truth is randomly shuffled. It can be seen that every model can generate images that depend on the conditions in terms of prediction performance. PUNet and VUNet achieved higher prediction performance than the Real data and FUNS in both CelebA and CHC. This may be because PUNet and VUNet generated less diverse images than the Real data and FUNS, which can be seen in $c$-LPIPS measure in Table 1.

Condition-specific and condition-invariant diverse samples, $x^{\text{sp}}$ and $x^{\text{iv}}$, respectively, generated by FUNS following Algorithm 1 are shown in Fig. 6. In CelebA, it is interesting to note that diverse Smiling images of the same individual were sampled as $x^{\text{sp}}$ when a Smiling image was given (in the

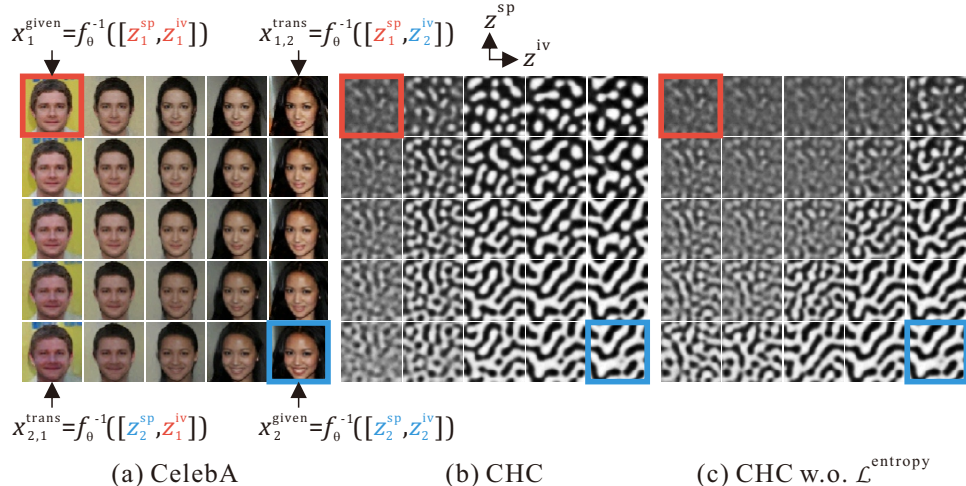

$x_1^{\text{given}} = f_\theta^{-1}([z_1^{\text{sp}}, z_1^{\text{iv}}])$    $x_{1,2}^{\text{trans}} = f_\theta^{-1}([z_1^{\text{sp}}, z_2^{\text{iv}}])$

$x_{2,1}^{\text{trans}} = f_\theta^{-1}([z_2^{\text{sp}}, z_1^{\text{iv}}])$    $x_2^{\text{given}} = f_\theta^{-1}([z_2^{\text{sp}}, z_2^{\text{iv}}])$

(a) CelebA                    (b) CHC                    (c) CHC w.o. $\mathcal{L}^{\text{entropy}}$

Figure 7: Feature disentanglement and interpolation. The top-left (red framed) and bottom-right (blue framed) images are given images, whereas the bottom-left and top-right images are style transferred images, $x^{\text{trans}}$. All of the other images are obtained by linear interpolation in latent space.

first column), whereas almost the same images were sampled as $x^{\text{sp}}$ when a non-Smiling image was given (the second column). This indicates that even if an individual is fixed, Smiling images have a certain amount of diversity, e.g., slightly grinning or, widely smiling, whereas non-Smiling images have less diversity. In contrast, diverse individual images of similar facial expressions were sampled as $x^{\text{iv}}$. This is because the diversity between individuals is almost independent of the condition of Smiling/non-Smiling. Therefore, the features corresponding to such diversity were embedded into $z^{\text{iv}}$. In the CHC dataset, the diversity of $x$ originates from both the thermal fluctuation and the elapsed time. As time proceeds, the black and white regions gradually separate and the characteristic length of their pattern grows. In Fig. 6, it can be seen that the elapsed time is captured by $x^{\text{iv}}$, whereas other fluctuation effects are captured in $x^{\text{sp}}$ because the characteristic lengths in $x^{\text{sp}}$ are similar to each other. The benefit of our model is that it can capture condition-specific diversity (if it exists) and can extract a feature that is truly independent of conditions.

An interpolation of two given images is shown in Fig. 7. Note that these results are obtained by using only two (top-left and bottom-right) images for each dataset. The remaining corners, top-right and bottom-left images, are obtained by exchanging the $z^{\text{iv}}$ of two given images. The vertical axis of Fig. 7 exhibits cross-conditional diversity because we can choose any two images for the top-left and bottom-right images whether they belong to the same conditions or not (the reader should not confuse this with condition-specific diversity). Figure 7 shows that the obtained image manifold is not only smooth but is aligned along meaningful axes, such as the individual and facial expression axes for CelebA (Fig. 7(a)) and the initial condition and elapsed time axes for CHC (Fig. 7(b)). We argue that the axis decomposition fails if $\mathcal{L}^{\text{entropy}}$ is absent (Fig. 7(c)). The effects of each loss term (Eq. (4)-(7)) are summarized in Table S.3 in the Supplementary Material.

## 6   Conclusion

Herein, we presented a framework for diverse image-to-image translation with feature disentanglement that is based on flow. Quantitative and qualitative comparisons showed that our model outperforms the state-of-the-art variational generative model in terms of image quality and image diversity. Furthermore, our model not only successfully generates diverse images but also can separate latent features into condition-specific and condition-invariant parts. By utilizing this property, we show that the meaningful orthogonal axes that lie in the latent space can be extracted by the given images.

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
