[Supplementary Material]

# Supplementary Material for "Flow-based Image-to-Image Translation with Feature Disentanglement"

## A Reformulation of the loss functions

### A.1 FUNS

The loss function for flow can be reformulated as follows.

$$
\begin{aligned}
\mathcal{L}^{\text{flow}} &= \mathbb{E}_{x,c\sim p(x,c)}\Big[ -\log e_\phi(f_\theta(x)|c) - \sum_p \log J_p(x)\Big] \\
&= \mathbb{E}_{x,c\sim p(x,c)}\big[ -\log p_{\theta,\phi}(x|c)\big],
\end{aligned}
\tag{S.1}
$$

where $p_{\theta,\phi}(x|c) := e_\phi(f_\theta^{-1}(z)|c)$, $x = f_\theta^{-1}(z)$.

Our model assumes that $p(z|c) := e_\phi(z|c)$ $(= \mathcal{N}(z|\mu_\phi(c), \operatorname{diag} \sigma_\phi^2(c)))$. Then the following holds for any conditional distribution $d_\psi(c|z)$:

$$
\begin{aligned}
-I(c;z) &= -\int p(c,z)\log\frac{p(c,z)}{p(c)p(z)}\,dcdz \\
&= -\int p(c,z)\log\frac{d_\psi(c|z)}{p(c)}\frac{p(c|z)}{d_\psi(c|z)}\,dcdz \\
&= -\int p(c)e_\phi(z|c)\log d_\psi(c|z)\,dcdz + \int e_\phi(z|c)p(c)\log p(c)\,dcdz \\
&\quad - \int p(z)p(c|z)\log\frac{p(c|z)}{d_\psi(c|z)}\,dcdz \\
&= \mathbb{E}_{c\sim p(c),\ z\sim e_\phi(z|c)}\big[ -\log d_\psi(c|z)\big] - \mathcal{H}\big[p(c)\big] - \mathbb{E}_{z\sim p(z)}\big[D_{\text{KL}}\big(p(c|z)\|d_\psi(c|z)\big)\big] \\
&= \mathcal{L}^{\text{recons}} - \mathcal{H}\big[p(c)\big] - \mathbb{E}_{z\sim p(z)}\big[D_{\text{KL}}\big(p(c|z)\|d_\psi(c|z)\big)\big] \\
&\leq \mathcal{L}^{\text{recons}} - \mathcal{H}\big[p(c)\big],
\end{aligned}
\tag{S.2}
$$

where $\mathcal{L}^{\text{recons}}$ is given by

$$
\mathcal{L}^{\text{recons}} = \mathbb{E}_{c\sim p(c),\ z\sim e_\phi(z|c)}\big[ -\log d_\psi(c|z)\big].
\tag{S.3}
$$

From the fact that $\mathcal{H}\big[p(c)\big]$ is constant, we can increase $I(c;z)$ by decreasing $\mathcal{L}^{\text{recons}}$.

Consequently, a loss function of FUNS without regularization terms can be written as

$$
\begin{aligned}
\mathcal{L}^{\text{flow}} + \mathcal{L}^{\text{recons}} = {}& \mathbb{E}_{x,c\sim p(x,c)}\Big[ -\log p_{\theta,\phi}(x|c)\Big] - I(c;z) \\
& + \mathbb{E}_{z\sim e_\phi(z)}\big[D_{\text{KL}}\big(e_\phi(c|z)\|d_\psi(c|z)\big)\big] + \mathcal{H}\big[p(c)\big]
\end{aligned}
\tag{S.4}
$$

In the above equation, it can be seen that $\mathcal{L}^{\text{flow}} + \mathcal{L}^{\text{recons}}$ is an upper bound of the objectives for conditional generation, which can be expressed as follows:

$$\mathcal{L}^{\text{flow}} + \mathcal{L}^{\text{recons}} - \underbrace{\mathcal{H}\big[p(c)\big]}_{\text{constant}} \geq \mathbb{E}_{x,c\sim p(x,c)}\Big[ -\log p_{\theta,\phi}(x|c)\Big] - I(c;z) \tag{S.5}$$

## A.2   CVAE

Let $q_\eta$ be an approximation of $p_\theta$. The loss function of the conditional variational autoencoder (CVAE) can be reformulated as follows.

$$\mathcal{L}^{\text{CVAE}}(c,x) = \mathbb{E}_{z\sim q_\eta(z|c,x)}\Big[ -\log p_\theta(x|c,z) - \log p_\theta(c) - \log p_\theta(z) + \log q_\eta(z|c,x)\Big]$$

$$= \mathbb{E}_{z\sim q_\eta(z|c,x)}\Big[ -\log p_\theta(x|c,z) - \log p_\theta(c) - \log p_\theta(z) + \log q_\eta(z|c,x)$$
$$+ \log p_\theta(c,x) - \log p_\theta(c,x)\Big]$$

$$= -\log p_\theta(c,x) + \mathbb{E}_{z\sim q_\eta(z|c,x)}\left[ \log q_\eta(z|c,x) - \log \frac{p_\theta(c)p_\theta(z)p_\theta(x|c,z)}{p_\theta(c,x)}\right]$$

$$= -\log p_\theta(c,x) + \mathbb{E}_{z\sim q_\eta(z|c,x)}\left[ \log q_\eta(z|c,x) - \log p_\theta(z|c,x) - \log \frac{p_\theta(c)p_\theta(z)}{p_\theta(c,z)}\right]$$

$$= -\log p_\theta(c,x) + D_{\text{KL}}\big(q_\eta(z|c,x)\|p_\theta(z|c,x)\big) + \mathbb{E}_{z\sim q_\eta(z|c,x)}\left[ \log \frac{p_\theta(c,z)}{p_\theta(c)p_\theta(z)}\right]$$

Here we assume that $p_\theta(c) = p(c)$. The Expectation of $\mathcal{L}^{\text{CVAE}}(c,x)$ can be written as

$$\mathbb{E}_{c,x\sim p(c,x)}\big[\mathcal{L}^{\text{CVAE}}(c,x)\big] = \mathbb{E}_{c,x\sim p(c,x)}\big[ -\log p_\theta(x|c) + D_{\text{KL}}\big(q_\eta(z|c,x)\|p_\theta(z|c,x)\big)\big]$$

$$+ \mathbb{E}_{z\sim q_\eta(z|c,x)}\left[ \log \frac{p_\theta(c,z)}{p_\theta(c)p_\theta(z)}\right] + \mathcal{H}\big[p(c)\big] \tag{S.6}$$

The last term becomes $I(c;z)$ when $p_\theta = q_\eta = p$.

## A.3   PUNet and VUNet

The loss function of the probabilistic U–Net (PUNet) can be reformulated as follows.

$$\mathcal{L}^{\text{PUNet}}(c,x) = \mathbb{E}_{z\sim q_\eta(z|c,x)}\Big[ -\log p_\theta(x|c,z)\Big] + D_{\text{KL}}\big(q_\eta(z|c,x)\|p_\theta(z|c)\big)\Big]$$

$$= \mathbb{E}_{z\sim q_\eta(z|c,x)}\Big[ -\log p_\theta(x|c,z) + \log q_\eta(z|c,x) - \log p_\theta(z|c)$$
$$+ \log p_\theta(x|c) - \log p_\theta(x|c)\Big]$$

$$= -\log p_\theta(x|c) + \mathbb{E}_{z\sim q_\eta(z|c,x)}\left[ \log q_\eta(z|c,x) - \log \frac{p_\theta(x|c,z)p_\theta(z|c)}{p_\theta(x|c)}\right]$$

$$= -\log p_\theta(x|c) + \mathbb{E}_{z\sim q_\eta(z|c,x)}\Big[ \log q_\eta(z|c,x) - \log p_\theta(z|c,x)\Big]$$

$$= -\log p_\theta(x|c) + D_{\mathrm{KL}}\big(q_\eta(z|c,x)\|p_\theta(z|c,x)\big)$$

Here, we assume the coefficient of the Kullback–Leibler divergence, $\beta$, which appeared in an original article [2], is equal to 1. The expectation of $\mathcal{L}^{\mathrm{PUNet}}(c,x)$ can be written as

$$\mathbb{E}_{c,x\sim p(c,x)}\big[\mathcal{L}^{\mathrm{PUNet}}(c,x)\big] = \mathbb{E}_{c,x\sim p(c,x)}\big[-\log p_\theta(x|c) + D_{\mathrm{KL}}\big(q_\eta(z|c,x)\|p_\theta(z|c,x)\big)\big] \quad (\mathrm{S.7})$$

Variational U–Net [1] takes a similar form of loss function except that it uses perceptual loss for the reconstruction.

### A.4 Comparison of the loss functions

Comparing Eq. (S.4)), (S.6)) and (S.7), we find that the first terms are common for all models. The main difference is that FUNS does not include approximations of $p_\theta$ ($q_\eta$), whereas both CVAE and PUNet include it. In addition, the mutual information terms that appear in FUNS and CVAE are different. Our model tends to increase $I(c;z)$ while CVAE tends to decrease $I(c;z)$. In CVAE, the condition–specific and condition–invariant parts are explicitly separated in the model architecture. To completely disentangle these parts, which are written as $c$ and $z$ in the above equation, respectively, CVAE needs to decrease $I(c;z)$. On the contrary, in our model, the condition–specific and condition–invariant parts of $z$ are not explicitly separated in advance. Instead, our model includes squeeze modules that act to decrease $I(c;z^{\mathrm{iv}})$. To preserve an important feature of $c$ during encoding, an additional term that increases $I(c;z^{\mathrm{sp}})$ is required. This is compensated for by $\mathcal{L}^{\mathrm{recons}}$. In PUNet, $z$ is fully conditioned on $c$. Therefore, there is no cost for $I(c;z)$.

## B  FID and LPIPS

Table S.1: Comparison of FID, LPIPS and $c$–LPIPS for CelebA (means and standard deviations of five trials)

|  | CelebA | | |
|---|---|---|---|
|  | FID | LPIPS | $c$–LPIPS |
| VUNet $_{T=1.0}$ | $66.000 \pm 4.328$ | $0.148 \pm 0.010$ | $0.146 \pm 0.009$ |
| VUNet $_{T=0.8}$ | $81.665 \pm 3.700$ | $0.105 \pm 0.007$ | $0.103 \pm 0.006$ |
| PUNet $_{T=1.0}$ | $114.760 \pm 9.172$ | $0.182 \pm 0.021$ | $0.180 \pm 0.021$ |
| PUNet $_{T=0.8}$ | $117.211 \pm 5.474$ | $0.149 \pm 0.018$ | $0.146 \pm 0.018$ |
| FUNS $_{T=1.0}$ | $39.561 \pm 3.898$ | $\mathbf{0.264 \pm 0.002}$ | $\mathbf{0.262 \pm 0.002}$ |
| FUNS $_{T=0.8}$ | $\mathbf{29.497 \pm 3.467}$ | $0.259 \pm 0.013$ | $0.256 \pm 0.013$ |
| Real data | – | $0.286$ | $0.284$ |

Table S.2: Comparison of FID, LPIPS and $c$–LPIPS for CHC (means and standard deviations of five trials)

|  | CHC | | |
|---|---|---|---|
|  | FID | LPIPS | $c$–LPIPS |
| VUNet $_{T=1.0}$ | $96.523 \pm 2.087$ | $0.217 \pm 0.007$ | $0.118 \pm 0.004$ |
| VUNet $_{T=0.8}$ | $163.968 \pm 5.678$ | $\mathbf{0.249 \pm 0.007}$ | $0.113 \pm 0.003$ |
| PUNet $_{T=1.0}$ | $225.687 \pm 6.116$ | $0.226 \pm 0.007$ | $0.108 \pm 0.003$ |
| PUNet $_{T=0.8}$ | $227.924 \pm 6.027$ | $0.214 \pm 0.008$ | $0.088 \pm 0.005$ |
| FUNS $_{T=1.0}$ | $\mathbf{10.480 \pm 2.048}$ | $0.207 \pm 0.010$ | $\mathbf{0.157 \pm 0.006}$ |
| FUNS $_{T=0.8}$ | $11.127 \pm 2.496$ | $0.210 \pm 0.011$ | $0.155 \pm 0.007$ |
| Real data | – | $0.225$ | $0.169$ |

## C  Abblation study

Table S.3: Effects of each loss term on the prediction accuracy of the conditions and dimensionality of $z^{\mathrm{sp}}$.

|  | CelebA | | CHC | |
|---|---|---|---|---|
|  | Acc. | $\dim(z^{\mathrm{sp}})$ | Err. | $\dim(z^{\mathrm{sp}})$ |
| $\mathcal{L}^{\mathrm{flow}}$ | $0.956$ | $5,525$ | $9.92$ | $2,062$ |
| $+\mathcal{L}^{\mathrm{recons}}$ | $0.981$ | $5,228$ | $9.62$ | $1,977$ |
| $+\mathcal{L}^{\mathrm{squeeze}}$ | $0.978$ | $86$ | $9.72$ | $270$ |
| $+\mathcal{L}^{\mathrm{entropy}}$ | $0.977$ | $113$ | $9.67$ | $111$ |

Table S.3 shows the results of the ablation study that was carried out to show the effect of each loss term. First, FUNS was trained only with $\mathcal{L}^{\mathrm{flow}}$. Then, other loss terms were added sequentially. The results show that the dimensionality of $z^{\mathrm{sp}}$, which is equivalent to the number of non-zero elements in $M$, significantly decreases when $\mathcal{L}^{\mathrm{squeeze}}$ and $\mathcal{L}^{\mathrm{entropy}}$ are added. In contrast, $\mathcal{L}^{\mathrm{recons}}$ has almost no contribution to decreasing $\dim(z^{\mathrm{sp}})$. Finally $\mathcal{L}^{\mathrm{recons}}$ slightly improve Acc. and Err., which is because the $\mathcal{L}^{\mathrm{recons}}$ term helps increas $I(c; z)$, which is the mutual information between $c$ and $z$, as shown in Eq. (S.5).

# D Samples

Figure S.1: Generated samples of CelebA. Non-Smiling images are shown in the first five rows and Smiling images are shown in the last five rows (FUNS, $T = 0.8$).

Figure S.2: Generated samples of CelebA. Non-Smiling images are shown in the first five rows and Smiling images are shown in the last five rows (PUNet, $T = 1.0$).

Figure S.3: Generated samples of CelebA. Non-Smiling images are shown in the first five rows and Smiling images are shown in the last five rows (VUNet, $T = 1.0$).

Figure S.4: Real samples from CHC for each condition. The images in the first column are the conditions.

Figure S.5: Generated samples for CHC for each condition (FUNS, $T = 1.0$). The images in the first column are the conditions.

Figure S.6: Generated samples for CHC for each condition (PUNet, $T = 1.0$). The images in the first column are the conditions.

Figure S.7: Generated samples for CHC for each condition (VUNet, $T = 1.0$). The images in the first column are the conditions.

(a) $x^{\mathrm{iv}}$

(b) $x^{\mathrm{sp}}$

Figure S.8: Additional $x^{\mathrm{iv}}$ and $x^{\mathrm{sp}}$ for CelebA with a Smiling given image (FUNS, $T = 0.8$).

(a) $x^{\text{iv}}$

(b) $x^{\text{sp}}$

Figure S.9: Additional $x^{\text{iv}}$ and $x^{\text{sp}}$ for CelebA with a non-Smiling given image (FUNS, $T = 0.8$).

(a) $x^{\text{iv}}$

(b) $x^{\text{sp}}$

Figure S.10: Additional $x^{\text{iv}}$ and $x^{\text{sp}}$ for CHC (FUNS, $T = 1.0$).

Figure S.11: Additional interpolation result. The top-left and bottom-right are the given images (test images).

Figure S.12: Failure cases for interpolation. The top-left and bottom-right are the given images (test images).