[Reviews · NeurIPS 2019]

Reviewer 1



Original: the paper is original in the way it uses flow based generative models for 1-to-many image to image translation. The recent paper [21] also uses conditional flow based models, although there are differences. Quality: The proposed approach is technically sound, and supported by good results in the experimental section. However it would have been nice to add into the comparisons the work of [21]. Was this omitted due to how recent that paper is? Also the way conditions were introduced into the CelebA dataset seems quite artificial: if only one couple of images were selected as prototypes for similing and not smiling, there might be many other differences between them that the model may pick up, and not only that of smiling. Which network was used to compute the FID in the CHC dataset? One last point in this regard, \mu(c)_i is supposed to be 0 when M_i is 0, but this is only so if b_\mu is also 0 (same for \sigma). Are the biases terms not used in practise? Clarity: the paper is overall well written, and the loss functions are well motivated. Some parts could be made clearer, particularly when going through eq. 1 and 2, there should be a reference to Fig 2c. I would make Fig 2c a new figure on its own, as IMHO it is the diagram that provides the most information about the approach. Significance: results look clearly better than competitors, although in an unconventional set of experiments. Still it may become a reference for other researchers working in similar problems. It also provides a new instance of that problem (CHC), which may be useful for the community. Edit after rebuttal: Having read the other reviews, and given that the rebuttal addressed my concerns (except that regarding the use of Inception-V3 to calculate FID for CHC, as CHC images do not look anything like natural images used for pretraining Inception-V3), I agree this paper should be published, and hence I raise my assessment from 6 to 7.

Reviewer 2



Overall, I like the idea and the task is also interesting. - For CelebA dataset, the authors choose “smile” or “not smile”. How is the other attribute cases such as young-aged or blond-black? - As the metric, classification accuracy for condition might be a metric for comparison. - As the authors said, a flow-based model such as Glow can generate high-quality images. But the generated face images are difficult to be high-quality. Could FUNS deal with higher-resolution datasets such as CelebHQ? If not, what is the main advantage of the FUNS against GAN or VAE-based I2I models? - Ablation study is required with respect to losses and modules. Minor Liu et al. [1] is a Flow-based I2I model performing conditional image generation. Even if [1] was published after NeurIPS deadline, I recommend revising the Introduction because the authors claimed their work is the first flow-based I2I in the Introduction. This work was referred in Section 3.1. Line 96 in page 3, merginalizing → marginalizing [1] Liu et al. Conditional Adversarial Generative Flow for Controllable Image Synthesis. CVPR 2019. [After rebuttal] I carefully read the other reviewers' comments and author feedback. The authors alleviate most of my concerns. Even if the classification accuracy is not competitive, the overall performance looks promising, considering the diversity from LPIPS score. Therefore, I update my score to 7.

Reviewer 3



Flows maximize the likelihood of observed images in a latent space and are formulated invertibly, such that at test time, latent space samples can be decoded into images. In their vanilla formulation, their setup does not allow to to condition on external attributed or images. This submission proposes an adaptation of flows for image-to-image translation problems and as such is an interesting and original work. The paper is overall well structured. The general idea is somewhat hard to understand until Section 3.3 / 3.4. This is also in part due to the complex nature of Figures 2a) -2c), which don’t aid much in clarifying the idea that underpins the model. It would be beneficial to state the basic idea in simple terms early on in the paper. I would have liked to read something like `The model consists of a U-Net with multiple scales of latent variables between its encoder and its decoder. These latent spaces encode an image c to condition on, which is trained by reconstructing c. Simultaneously a flow based model is trained to associate possible outputs x with the given image c by maximizing the likelihood of the latents z that the flow produces given x under the U-Net’s latent space densities for c’. Training the encoder-decoder is done using 3 terms, a L2-reconstruction term on the output space, an L1-term to encourage a sparse encoding of z|c by means of a mask that yields Normal and thus uninformative latent distributions and an entropy term on z|c that encourages the `unmasked’ latent distributions’ to have low variance. This setup seems a bit contrived and could potentially be avoided / done in a more principled way by training the encoder-decoder via variational inference? For improved clarity of the training objectives, the loss terms (left-hand sides of Eqs. 4, 5 and 6) should have the parameters with respect to which they are optimized as a subscript. It would further be interesting to have an analysis of how many of the latents end up being masked (surely a function of the weighting terms). Also, having latents at all scales, it appears the encoder has an incentive to go the easiest route and encode as much as possible in the scale with the highest resolution, which thus wouldn’t require finding a more abstract and semantically structured latent space? In this context the means (\mu) of e(z|c) of individual encoder scales could be fixed to investigate what individual scales encode, in case it can be teased apart. It would generally be helpful to give the dimensions of the latent tensors at all scales, to better understand how the dimensionality d_x is split across scales. The employed metrics test for the fidelity and the diversity of the generated images only, but do not seem to test for whether they are appropriate / plausible, given the image to condition on. Other datasets allow to do so more easily, e.g. image translation between street scene images on Cityscapes and its segmentation, which was considered by other image-to-image translation works. A way to quantify conditional plausibility on CelebA could be to pretrain a classifier to classifiy the ~40 attributes of each image and use this classifier to quantify whether they are conserved / altered in the generated images. This seemsan important analysis given that the proposed model does not have a deterministic path from c -> x, which means there could potentially be a very weak correlation only. Additional the likelihood of ground truth images x under the encoder could be reported, so e(f(x)|c). The PU-Net is reported to produce blurry results, but it has not been stated what the exact architectural setup and training procedure for this baseline (and for the VU-Net) was. There are various typos and errors in grammar such as `merginalizing’, missing white spaces, wrong/missing articles (`a’, `the’), `there is no oracle that [?] how large’, wrong usage of colons in datasets description. [After Rebuttal] The rebuttal addressed the comments and questions I had raised. My score remains at 7.

[Author Response · NeurIPS 2019]

We thank the reviewers for fruitful comments. Here, we respond to the major comments. For the minor points like
presentation issues, we will fix them in the camera-ready (and *we do not mention them here due to space limitation.*)

**Reviewer #1** (i) We could not compare our method to [21], because [21] was published in this April and their
code was not available. (ii) As the reviewer mentioned, a couple of images, say $c_{\text{smile}}$ and $c_{\text{non-smile}}$, were used as
conditions for smiling and non-smiling images, respectively. Our model is trained using $\{(x, c_{\text{smile}}) \mid x \in X_{\text{smile}}\}$ and
$\{(x, c_{\text{non-smile}}) \mid x \in X_{\text{non-smile}}\}$, where $X_{\text{smile}}$ and $X_{\text{non-smile}}$ are sets consisting of smiling and not-smiling images in
CelebA, respectively. Our model then learns the difference between $x \in X_{\text{smile}}$ and $x \in X_{\text{non-smile}}$ in the embedded
space, in which $c_{\text{smile}}$ and $c_{\text{not-smile}}$ are only used to distinguish whether $x \in X_{\text{smile}}$ or $x \in X_{\text{non-smile}}$. In contrast, our
CHC experiment demonstrates a situation in which the condition image (i.e., the initial microstructure) of each image
has more diversity. (iii) We used Inception-V3 to calculate FID for CHC. (iv) We apologize that the description "make
a part of $z$ follow the non-informative distribution $\mathcal{N}(0, 1)$" in line 125 of our submitted manuscript was incorrect. It
should be modified as "make a part of $z$ follow $\mathcal{N}(f_\mu(b_\mu), \text{diag}(\exp f_\sigma(b_\sigma)))$ that is independent of $c$".

**Reviewer #2** (i) We trained FUNS using CelebA with Male-Female conditions. An interpolated
result is shown in Fig. A. We can see that $z$ is successfully disentangled into the condition
dependent ($z^{\text{sp}}$) and independent ($z^{\text{iv}}$) parts.

Fig. A

(ii) We trained a ResNet to predict smile/non-smile for CelebA, and then, evaluated the prediction
accuracy for images generated by FUNS, PUNet, and VUNet, as well as the real test set of CelebA.
For CHC, we trained in-house U-Net to predict the initial condition, and then, evaluated the
prediction error using $L_2$ distance between the ground truth and the prediction. The results are
summarized in Table A. Note that the prediction error for the CHC prediction is 22.84 when the
ground truth are randomly shuffled. We see that every model can generate images that depend on the conditions in
terms of the prediction accuracy. PUNet and VUNet achieved higher prediction accuracy compared to Real/FUNS in
both CelebA and CHC. This may be because PUNet and VUNet generated similar images that are easy to predict the
condition. As the c-LPIPS scores suggest, PUNet and VUNet generated less diverse images compared to Real/FUNS.

(iii) We have not tried to train FUNS with CelebA-HQ. It is uncertain if FUNS can
be successfully trained using HQ images (further techniques might be required).
However, as shown in Table 1 of our paper, FUNS has advantages over VAE-based
I2I models (incl. PUNet, VUNet) in the image quality (FID) and diversity (LPIPS).
The main advantage of flow-based models over GAN-based ones is, as discussed
in [14], that they have invertible mappings between images and latent codes, which
will be useful for downstream tasks, e.g., Gómez-Bombarelli, R., et al., *ACS Cent. Sci.*, **4**, (2018), 268–276.

| (Table A) | CelebA | | CHC | |
|---|---|---|---|---|
| | Acc. | c-LPIPS | Err. | c-LPIPS |
| Real | 0.924 | 0.284 | 9.54 | 0.169 |
| FUNS | 0.973 | 0.262 | 9.67 | 0.157 |
| PUNet | 1.000 | 0.180 | 9.48 | 0.108 |
| VUNet | 0.977 | 0.146 | 9.46 | 0.118 |

(iv) We carried out an ablation study to show the effect of each loss term (only with CHC
dataset due to time limitation). We first trained FUNS using only $\mathcal{L}^{\text{flow}}$, and then, other
loss terms are added sequentially. The results are shown in Table B, in which $L_2$ errors
(explained in Table A) and the number of non-zero elements in $M$ ($\dim(z^{\text{sp}})$) are reported.
The lower error means that the generated images are more related to the respective conditions.

| (Table B) | Err. | $\dim(z^{\text{sp}})$ |
|---|---|---|
| $\mathcal{L}^{\text{flow}}$ | 9.56 | 4,096 |
| $+\mathcal{L}^{\text{recons}}$ | 9.51 | 4,040 |
| $+\mathcal{L}^{\text{squeeze}}$ | 9.58 | 1,095 |
| $+\mathcal{L}^{\text{entropy}}$ | 9.67 | 550 |

From Table B, if we train FUNS by using only $\mathcal{L}^{\text{flow}}$, $\dim(z^{\text{sp}}) = 4096$, which is in fact equal to the whole latent
variable (therefore then $\dim(z^{\text{iv}}) = 0$), suggesting that all the latent variables are dependent on the condition $c$. In
contrast, by adding proposed loss terms, $\dim(z^{\text{sp}})$ decreases to 550 (then $\dim(z^{\text{iv}}) = 3546$), while the prediction error
is almost maintained. This result suggests that the input image $x$ is successfully disentangled in the latent space into
condition-dependent/independent parts. The impact of dropping $\mathcal{L}^{\text{entropy}}$ is also illustrated in Fig. 5 of our manuscript.

**Reviewer #3** (i) We modified our manuscript to clarify the basic idea in the earlier part
of the paper according to the reviewer's comment. (ii) As the reviewer pointed out, it
is interesting to apply variational inference to train the encoder-decorder in our model,
which will be described as a future work. (iii) As shown in Table B, 550 of latent
variables were unmasked (therefore the remaining 3546 latents were finally masked) for
CHC dataset. In more detail, let $z^{(l)}$ be the latent variables at level $l$ ($z^{(1)} \in \mathbb{R}^{32 \times 32 \times 2}$,
$z^{(2)} \in \mathbb{R}^{16 \times 16 \times 4}$, $z^{(3)} \in \mathbb{R}^{8 \times 8 \times 8}$, and $z^{(4)}, z^{(5)} \in \mathbb{R}^{4 \times 4 \times 16}$), the numbers of unmasked
latents were 2, 2, 200, 181, and 165 for $l = 1, \ldots, 5$, respectively. It means that the
latents in higher resolution layers are more likely to be masked. (iv) We generated

$z^{(1)}, z^{(2)}$ and $z^{(3)}$ are sampled

$z^{(4)}$ is sampled

$z^{(5)}$ is sampled

Fig. B

CelebA samples using FUNS with the condition of Smile by varying only a part of $z^{(l)}$, while the remaining $z^{(l)}$ are
fixed to $f_\mu(b_\mu^{(l)})$. Fig. B shows the results: (top) $z^{(1)}$, $z^{(2)}$, and $z^{(3)}$ are sampled. (middle) $z^{(4)}$ is sampled. (bottom)
$z^{(5)}$ is sampled. We can see that larger diversity is captured by latent variables in lower resolution layers ($z^{(4)}$ and $z^{(5)}$).
Latents in high resolution layers ($z^{(1)}$, $z^{(2)}$, and $z^{(3)}$) control very subtle facial expressions. (v) Please refer to response
(ii) to Reviewer 2 for the experimental results that evaluate the proposed model in terms of the prediction accuracy by
pretrained classifiers. (vi) We will correct grammatical issues through an English proofreading service.

[Meta-Review · NeurIPS 2019]

I thank the authors for their submission. The paper presents a flow-based generative model to perform 1-to-many image to image translation. I strongly encourage the authors to take into account the reviewers' comments and concerns for the final manuscript.